# Radiomics using disentangled latent features from deep representation learning in soft-tissue sarcoma

**Timothy Sum Hon Mun**[1]                    TIMOTHY.SUMHONMUN@ICR.AC.UK
**Amani Arthur**[1]                    AMANI.ARTHUR@ICR.AC.UK
**Imogen Thrussell**[1]                    IMOGEN.THRUSSELL@ICR.AC.UK
**Jessica Winfield**[1,2]                    JESSICA.WINFIELD@ICR.AC.UK
**Dow-Mu Koh**[1,2]                    DOW-MU.KOH@ICR.AC.UK
**Paul Huang**[1]                    PAUL.HUANG@ICR.AC.UK
**Christina Messiou**[1,2]                    CHRISTINA.MESSIOU@RMH.NHS.UK
**Simon J Doran**[1]                    SIMON.DORAN@ICR.COM
**Matthew D Blackledge** [1]                    MATTHEW.BLACKLEDGE@ICR.AC.UK
[1] *Institute of Cancer Research, London, United Kingdom*
[2] *Royal Marsden NHS Foundation Trust, London, United Kingdom*

## Abstract

Detecting the treatment response of radiotherapy for rare cancers such as soft-tissue sarcomas (STS) is difficult due to the intratumoral heterogeneity of the disease within tumors. STS are a group of diseases with more than 70 recognized subtypes with each of them having distinctive histological and clinical-pathological characteristics. Apparent Diffusion Coefficient (ADC) mapping provides a quantitative measure of the magnitude of water diffusion in biological tissues which can provide insight into the microstructure of tissues. An unsupervised deep representation learning pipeline that can learn disentangled and interpretable radiomics features from the Apparent Diffusion Coefficient (ADC) maps of patients has been developed and the learnt latent features have been assessed for baseline test-retest repeatability as well as for outcome prediction in a pilot cohort.

**Keywords:** MRI, Diffusion-Weighted Imaging, Radiomics, Deep Learning, Representation Learning

## 1. Introduction

Soft tissue sarcomas (STS) demonstrate intratumoral heterogeneity, making it difficult to successfully monitor response to treatment using conventional size-based criteria. Radiomics (Gillies et al., 2016) can offer opportunities to find novel biomarkers of treatment response by quantifying the level of intra-tumoral heterogeneity within tumours via the measurement of "hand-crafted" features that aim to represent tumour image statistics, shape, and texture.

A potential disadvantage of these features is that they are not necessarily data-driven and thus may miss characteristics within the image that could be important for demonstrating tumour response. Deep learning (Sum Hon Mun et al., 2022) has also been shown to be able to extract interesting features but it can be hard to interpret these features especially if they are extracted directly from the layers of convolutional neural networks (CNNs).

In this work, we explore the use of generative models known as Variational Autoencoders (VAEs) (Kingma and Welling, 2013) which learn a mapping between the original image

and latent feature representation that can accurately reconstruct the image. We apply this approach to maps of apparent diffusion coefficient (ADC) as derived from diffusion-weighted MR-imaging of a pilot cohort of patients with retroperitoneal soft-tissue sarcoma (STS). To determine the potential sensitivity of the derived latent features to potential post-therapeutic change, we investigate their test-retest repeatability. We subsequently evaluate their predictive potential in a multivariate regression model of recurrence-free survival when combined with the patient demographic features.

## 2. Data and Methods

**Dataset Description**: Baseline and repeat baseline scans for 22 patients were acquired using axial diffusion-weighted imaging (DWI) (b = 50,600,900 s/mm2); ADC maps were extracted from these images using a least-squares monoexponential fit. A single radiologist outlined tumour regions-of-interest (ROIs) on T2-weighted images, which were subsequently transferred to the ADC maps. Augmentation techniques including rotation, translation, scaling, and flipping were performed independently for each slices to generate 2313 images in total from baseline scans, split into 2082 training and 231 validation. Input data consisted of two channels: (i) the complete ADC map slice, and (ii) the ADC map slice masked by the tumour ROI. The second channel allowed the network to focus on the features that represent the tumour region, whilst the first provided information about tissue surrounding the tumour. Both channels were resized using bilinear interpolation to 64 x 64.

  **Model**: Our VAE model architecture consisted of a 3-layer, 2D convolution encoder/decoder with a kernel size of three with the following channels [16, 32, 64] (encoder) $\rightarrow$ 7 (latent features) $\rightarrow$ [64, 32, 16, 2] (decoder).

  We used the beta-VAE (Higgins et al., 2017) variant to tune emphasis on the Kullback-Leibler divergence loss of the features ($\beta = 0.5$), using the following parameters: learning rate = 0.0001, epochs = 1000, Adam optimizer. After training, we evaluate the encoded features derived from the middle tumour slice of each patient (containing the largest tumour cross-section). As our goal is to derive a set of useful features, we train the model on the first baseline scans of all patients to capture as much information as possible given our small cohort.

  **Feature analysis**: Hierarchical agglomerative clustering on the pairwise Pearson correlation (r) between all extracted features from the first baseline scan identified linearly independent feature subgroups (independence was determined where $r2 > 0.5$). The intraclass correlation coefficient (ICC) was used to compare the repeatability of features (ICC = 1 indicating perfect repeatability, $ICC > 0.5$ indicating poor repeatability). Clinical features were combined with VAE features after normalization, and time to tumour recurrence was modelled using multivariate Cox proportional hazard models with elastic-net L1 penalty ratio = 0.9 and alpha = 0.17. The model was trained using 2-fold validation.

## 3. Results

**Feature analysis**: Bland-Altman (BA) plots in Figure 1a demonstrate excellent test-retest repeatability for all features (ICC = 0.86 – 0.98). These plots also demonstrate no systematic bias or any outliers in features generated from the second baseline measurements,

potentially providing further evidence that the VAE is not-overtraining. Furthermore, a correlation heatmap for baseline features demonstrated in Figure 1b indicates there are no clear correlations between any of the 7 features, suggesting that the VAE successfully enforces this important characteristic in the derived features.

**Recurrence-free survival analysis**: A feature importance plot is provided in 1c, summarising the coefficients for features in descending order; features 2 and 6 from the VAE rank highly alongside other clinical factors. To further demonstrate the predictive power of feature 6, we demonstrate survival curves of four patient groups (1d) obtained by discretized feature 6 into four groups with equal patient sizes. It is evident that the final bin (high value of feature 6) is associated with a higher risk of recurrence.

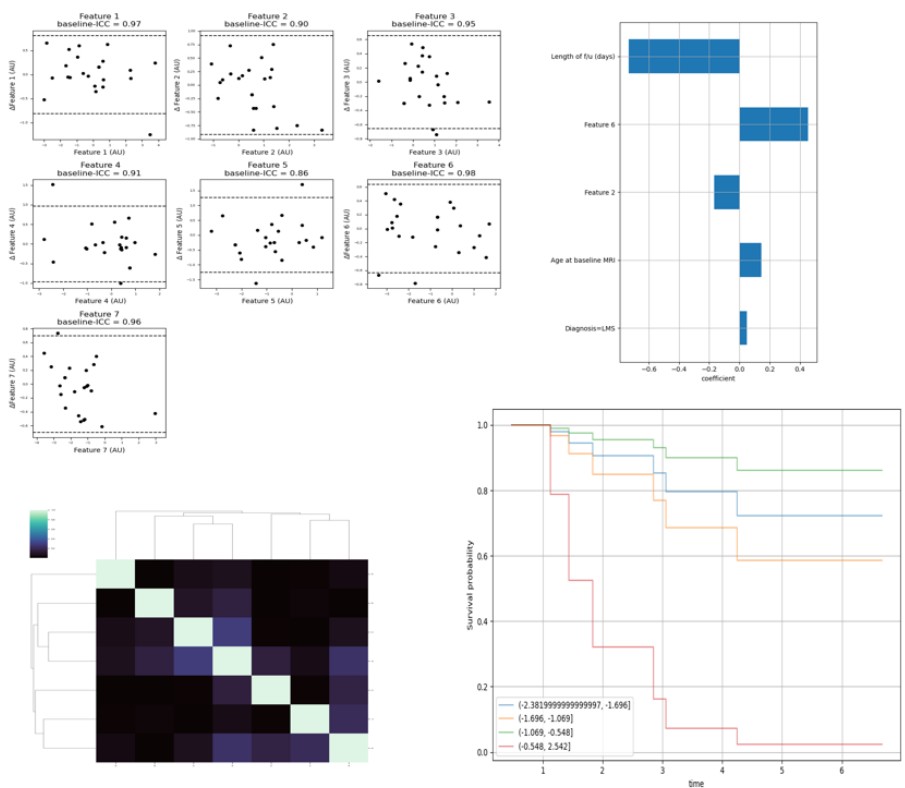

Figure 1: (a) Bland Altman plot (b) Correlation Heatmap (c) Feature importance (d) Kaplan Meier curve

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

## Acknowledgments

This work was supported by the International Accelerator Award funded by Cancer Research UK [C56167/A29363], Associazione Italiana per la Ricerca sul Cancro [AIRC - 24297] and Fundacion Cient 1 ca – Asociacion Espanola Contra el Cancer [Foundation AECC - GEACC19007MA]. We acknowledge Cancer Research UK and Engineering and Physical Sciences Research Council support to the Cancer Imaging Centre at Institute of Cancer Research and Royal Marsden Hospital in association with Medical Research Council and Department of Health C1060/A10334, C1060/A16464 and National Health Service funding to the National Institute for Health Research Biomedical Research Centre, Clinical Research Facility in Imaging and the Cancer Research Network

The report is independent research funded by the National Institute for Health Research. The views expressed in this publication are those of the author(s) and not necessarily those of the National Health Service, the National Institute for Health Research or the Department of Health. We also acknowledge the support of the Alan Turing Institute's Enrichment Scheme.

