# OpenReview forum: "Radiomics using disentangled latent features from deep representation learning in soft-tissue sarcoma"
_MIDL.io/2023/Short_Paper_Track — MIDL 2023 Short paper track Poster_

### Official Review · Reviewer_Bu9n · 2023-04-22
**Standard model applied to generation of radiomics features from ADC maps**

**Rating:** 6
**Confidence:** 4

**Review:**

This paper proposes an unsupervised learning approach to learn radiomics features for soft-tissue sarcomas. The apparent diffusion coefficient map and tumor ROI segmentation were used as input to a beta-VAE to learn a latent representation of the data. Training was performed on a pilot dataset of baseline patient images and repeatability of features was assessed using a repeat scan of the same patients. Finally, extracted latent image features were combined with clinical features to generate survival curves.

Strengths:
+ The unsupervised learning approach allows for latent representation learning on smaller dataset and without task annotations
+ An interesting test of repeatability is performed, where baseline scans were used for training, and similarity of extracted features in repeat scans compared to baseline scans was assessed.
+ Survival curves using important features suggest the usefulness of some learned imaging features

Weaknesses:
- Prior VAE model is used (beta-VAE), but not cited
- Unclear if image split was done patient-wise and exactly how validation data is used. Seems that training was just done on baseline scans but unclear if the 2313 images come from both baseline + rescan.
- Characterization of "outcome prediction" is a bit unfair - Unclear how time to recurrence tumor model was learned - seems to be using all the data, thus this is an analytical model fitting not predictive. Further suggested by survival analysis which appeared to be done using all the data split into 4 groups.

---

### Official Review · Reviewer_Mbis · 2023-04-24
**Radiomics using disentangled latent features from deep representation learning in soft-tissue sarcoma**

**Rating:** 5
**Confidence:** 3

**Review:**

This work uses variational auto encoders to learn features of soft-tissue sarcoma from MRI scans, and use these features in survival analysis showing patient stratification via Kaplan-Meier analysis.

PROS
The proposed approach seems to be learning meaningful features from raw data, which can be used to stratify patients with different prognosis.

CONS
I am confused by the main focus of the paper.
The title mentions radiomics, while instead a VAE is used, with unsupervised learning.
The way results are reported and discussed is a bit confusing for me, in the end, Kaplan-Meier curves are shown, it is unclear what variable is used and how cut-off values were computed; the correlation heat map refers to 8 features but it’s a 7x7 matrix.